# Evaluation of Vaccine Strains Developed for Efficient, Broad-Range Protection against Foot-and-Mouth Disease Type O

**DOI:** 10.3390/vaccines11020271

**Published:** 2023-01-27

**Authors:** Seong Yun Hwang, Sung Ho Shin, Hyun Mi Kim, SeHee Shin, Min Ja Lee, Su-Mi Kim, Jong-Soo Lee, Jong-Hyeon Park

**Affiliations:** 1Animal and Plant Quarantine Agency, 177 Hyeoksin 8-ro, Gimcheon 39660, Republic of Korea; 2College of Veterinary Medicine, Chungnam National University, Daejeon 34314, Republic of Korea

**Keywords:** foot-and-mouth disease, FMD, broad antigenic coverage, vaccine

## Abstract

Foot-and-mouth disease (FMD) type O includes 11 genetic topotypes. The Southeast Asia (SEA), Middle East–South Asia (ME-SA), and Cathay topotypes belong to FMD type O and occur frequently in Asia. Therefore, it is necessary to develop a potent vaccine strain with a broad antigenic coverage in order to provide complete protection against these three topotypes. In this study, an experimental vaccine was produced using chimeric vaccine strains (JC-VP1 or PA2-VP1) that contained VP4, VP2, and VP3 of the ME-SA topotype (O Manisa) and VP1 of the SEA topotype (Mya98 lineage; O/SKR/Jincheon/2014) or ME-SA topotype (PanAsia2 lineage; O/PAK/44). Mice were immunized with the experimental vaccines, and they were fully protected against the three topotypes. The neutralizing antibody titers of PA2-VP1 were significantly higher than those of JC-VP1 in the early vaccination phase in pigs. Here, we confirmed complete protection in pigs vaccinated with JC-VP1 or PA2-VP1, when challenged against the SEA (O/SKR/Jincheon/2014), ME-SA (O/SKR/Boeun/2017) and Cathay (O/Taiwan/97) topotype viruses, with moderately higher protection provided by PA2-VP1 than by JC-VP1.

## 1. Introduction

Foot-and-mouth disease (FMD) is a viral infectious disease that is highly contagious, and it induces various symptoms, such as fever, lameness, and vesicle formation, in several domestic animal species [1,2]. Unless an appropriate prevention system is in place at the beginning of an FMD outbreak, it can spread quickly and infect many animals, causing economic damage [3,4].

The FMD virus (FMDV) has seven serotypes, namely, O, A, Asia1, C, SAT1, SAT2, and SAT3, and several topotypes exist in each serotype. Many strains that belong to the same topotype often do not show cross-immunity, depending on their genetic lineages; therefore, a careful selection of vaccine strains is critical for achieving the appropriate protection and prevention of FMD [5]. The regional distribution of FMDV is classified into Pools 1–7 worldwide. Specifically, FMD caused by serotypes O, A, and Asia1, which belong to Pools 1–3, is continuing to cause an outbreak in Asia [6,7], such as in Vietnam, Thailand, and China. Recently, the type O Middle East–South Asia (ME-SA)/Ind2001 virus, which belongs to the Pool 2 region, was introduced into the Southeast and East Asia regions of the Pool 1 region and caused outbreaks [8,9,10]. As new viruses emerge due to continuous outbreaks that are difficult to control with existing vaccines, it has become very important to develop new vaccine strains that can provide protection against new viruses. As several mutated viruses coexist in areas where FMD is prevalent, a vaccine capable of providing a broader-range protection than the existing vaccines is required.

The Republic of Korea, located in East Asia, a nationwide vaccination policy has been implemented since 2010, using O Manisa that belongs to the ME-SA topotype [11]. In 2014, FMD outbreaks of the O (Southeast Asia) SEA topotype lasted for more than five months despite the implementation of the vaccine mandate policy. To countermeasure for this persisted outbreak, an additional vaccine containing O 3039 was urgently introduced as a booster shot, and as a result, the breakout was successfully controlled [12]. Hence, we now understand that there is an urgent need for an FMD vaccine that is effective in preventing various FMD virus types and that can provide protection in the case of an emergency FMD epidemic [11,13].

Ind2001, which belongs to the ME-SA topotype, occurs worldwide, whereas the Cathay topotype majorly circulates in Southeast Asia and China [14,15,16]. Hence, it is important to develop a broad-spectrum vaccine against FMDV type O, which is prevalent in Asia. Virus strains from previous outbreaks have been proliferated and modified to obtain recombinant vaccine strains that provide broad antigenic coverage [17]. However, securing various viruses from different topotypes and screening for a suitable virus strain that is replication-competent with a broad antigenicity coverage make it very difficult to succeed in developing a broad-spectrum vaccine strain [18]. Here, we created two chimeric antigens with the O Manisa template, JC-VP1 and PA2-VP1, from SEA and ME-SA viral genes, respectively, that could provide complete and wide-ranging protection against the three topotypes of FMDV type O, which is widespread in Asia. The effectiveness of the newly developed vaccine strains was examined both in vitro and in vivo.

## 2. Materials and Methods

### 2.1. Plasmid Preparation for Infectious Clones

Chimeric viruses were cloned, as described in a previous study [19]. The VP1 of the viral structural proteins was replaced by that of O1manisa amplified from a synthetic gene based on O/SKR/JC/2014 (O JC, GenBank No.KX162590.1) and O/PAK/44/2008 (O PA2, GenBank No.GU384682) using the following oligonucleotide primers:JC_VP1_F; 5′-ACCACTTCGACAGGCGAGTCG-3′ andJC_VP1_R; 5′- CTGCTTTACAGGTGCCACTAT-3′ for JC-VP1 cloningPA2_VP1_F; 5′-ACCACCTCCACAGGTGAGTCAG-3′ andPA2_VP1_R; 5′-CTGTTTCACAGGTGCCACTATC-3′ for PA2-VP1 cloning.

### 2.2. Cell Culture and Virus Recovery

ZZ-R 127 cells were maintained in Dulbecco’s Modified Eagle’s Medium/F12 (Corning, Union City, NJ, USA). BHK-21 cells were maintained in Dulbecco’s Modified Eagle’s Medium (Corning, Union City, NJ, USA) supplemented with 10% fetal bovine serum (Gibco BRL, Paisley, UK) and 1% antibiotic/antimycotic solution (Gibco). BHK-T7-9 cells were maintained in Glasgow Minimum Essential Medium (Gibco BRL, Paisley, UK) supplemented with 5% fetal bovine serum (Gibco BRL, Paisley, UK) and 10% tryptose phosphate broth (Sigma-Aldrich, St. Louis, MO, USA). BHK-21 suspension cells were maintained in a CD BHK-21 Production Medium (Gibco BRL, Paisley, UK). The cells were maintained in a 5% CO_2_ atmosphere at 37°C. The chimeric virus expression plasmids were linearized by the restriction enzyme *Spe*I (NEB, MA, USA), and the linearized plasmids were transfected in the BHK-T7-9 cells using Lipofectamine 3000 (Invitrogen, CA, USA). After incubation for 48–72 h at 37 °C, the virus was harvested in three freeze–thaw cycles, and the VP1-replaced viruses were amplified in fresh ZZ-R 127 cells. All FMDV-related experiments were performed in biosafety level 3 (BSL-3) at the Animal and Plant Quarantine Agency (APQA).

### 2.3. Virus Antigen Purification

The BHK-21 suspension cells were infected with the VP1-replaced virus and harvested via freezing and thawing when a complete cytopathic effect (CPE) was confirmed. The chimeric viruses were inactivated via treatment with 0.003 N binary ethylenimine for 24 h at 26 °C. The inactivated virus was subsequently precipitated with 7.5% PEG 6000 and 2.3% NaCl overnight at 4 °C and was concentrated 200 times with Tris-KCl buffer. The concentrated antigen was purified via centrifugation through a 15–45% sucrose gradient in Tris-KCl buffer at 30,500 rpm for 4 h in an SW41 rotor at 4 °C, and the fraction in which the antigen was present was determined via optical density measurements at 259 nm. The purified antigen was confirmed through visualization with a transmission electron microscope (Hitachi H7100FA, Tokyo, Japan).

### 2.4. Structural Modeling and Analysis

The crystal structure of O PanAsia (PDB accession no. 5NE4) was a template for predicting the JC-VP1 and PA2-VP1 capsid model using SWISS-MODEL. A comparative structure analysis of the protomeric subunit was performed using the Pymol molecular graphics system (v2.4.1, Schrodinger LLC, New York, NY, USA) [20].

### 2.5. Vaccination and Protection Evaluation in Adult C57BL/6 Mice

We used a rapid method in mice to evaluate the vaccines against various topotype viruses before carrying out challenge testing on target animals. C57BL/6 mice (6–7 weeks old females), supplied by the KOSA BIO Inc. (Gyeonggi, Republic of Korea), were used for this experiment (5 mice/group). The mice were managed in the Animal and Plant Quarantine Agency (APQA) and used with the approval of the Animal Care and Use Committee. The antigen was diluted in dose groups at 1/10, 1/40, 1/160, and 1/640 of the vaccination in pigs (15 μg/dose), and it was prepared with ISA 206 (double oil emulsion of water in oil in water, W/O/W type) and a 10% aluminum hydroxide gel adjuvant. The mice were vaccinated via intramuscular injections for 7 days, and they were challenged with the O/VIT/2013 (ME-SA topotype) virus following IP injection with 3 × 10^4.0^ TCID_50_/0.1 ml and the O/SKR/Jincheon/2014 (SEA topotype) and O/Taiwan/97 (Cathay topotype) viruses following the IP injection with 1 × 10^5.0^ TCID_50_/0.1 mL and observed for seven days.

### 2.6. The Immunogenicity of the Experimental Vaccine in Cattle and Pigs

The five-month-old cattle and 8- to 10-week-old pigs (n = 5) were inoculated with 15 μg/dose of antigen with ISA 206, saponin, and aluminum hydroxide gel adjuvant. The sera of the cattle were collected at 14, 28, 42, and 56 dpv, and the sera of the pigs were collected at 7, 14, 21, 28, 42, 56, and 84 dpv. The pigs were only boosted at 28 dpv. The antibodies to the structural proteins of FMDV in the serum were detected using a PrioCheck FMDV O (Prionics AG, Schlieren-Zurich, Switzerland). An animal was considered positive if the sample measured an inhibition value of >50%.

### 2.7. Virus Neutralization Test

The serum was heat-inactivated at 56 °C for 30 min. The cell density was adjusted to form a 70% monolayer, and 2-fold serial dilutions of samples were prepared. The diluted serum was incubated with FMDV 100 TCID_50_ of the virus for 1 h at 37 °C. LFBK (porcine kidney) cells were then added to every 96-well plate. CPE was checked after 2–3 days, and the titers were calculated as the log10 of the reverse antibody dilution required to neutralize 100 TCID_50_ of the virus. A titer of at least 1.65 (log10) is regarded as positive by the World Organisation for Animal Health (WOAH) [21].

### 2.8. Vaccine Matching Test

The two-dimensional virus neutralization test (2D-VNT) was conducted according to the foot-and-mouth disease manual [21]. Serum samples at 28 DPV were collected from 5 vaccinated cattle and pigs using the JC-VP1 and PA2-VP1 vaccines. The field viruses and homologous vaccine viruses were used for 2D-VNT. The neutralizing antibody titer of the vaccine serum against 100 TCID_50_ of each virus was estimated via regression. The r1 value was calculated as neutralizing antibody titer to field virus/neutralizing antibody titer to vaccine virus. An r1 value ≥ 0.3 was interpreted as cross-protected, and an r1 value < 0.3 was interpreted as unprotected.

### 2.9. Challenge Test of Immunized Pigs with Chimeric Vaccine

Three different viruses (each at 10^5.0^ TCID_50_/0.1 mL) were injected intradermally into the heel bulb for challenge testing at 28 dpv. Sera and oral swabs were collected for 8 days after the challenge. Blood samples were obtained via the anterior vena cava and collected into Vacutainer Serum Tubes (BD Biosciences, NJ, USA). For oral swabs, a BD™ Universal Viral Transport Kit (BD Biosciences, NJ, USA) was used. Clinical scores were calculated by summing the distributed scores using the following criteria: (a) hoof and foot vesicles (1–2 points per foot) and (b) snout, lips, and tongue vesicles (1 point for each area).

### 2.10. RT-PCR for Viremia Detection

After extracting the viral RNA from the serum and swab samples using QIAcube HT (QIAGEN, Valencia, CA, USA) according to the manufacturer’s protocol, a RT-PCR was performed using a one-step prime-script RT-PCR kit (Bioneer Inc., Daejeon, Republic of Korea) and a CFX96 TouchTM Real-Time PCR Detection System (Bio-Rad, CA, USA) for virus quantification.

### 2.11. Statistical Analyses

Data are presented as means ± standard deviations (SD) and represent the results from at least 3 independent experiments. The differences between groups were analyzed via an analysis of variance (ANOVA), and means were compared using Student’s t test. *p* values of ^∗^
*p* < 0.05 were regarded as significant or highly significant.

### 2.12. Ethics Statement

The animal experiments were performed in strict accordance with the recommendations of the guide for the care and use of laboratory animals of the Animal and Plant Quarantine Agency (APQA). All animal procedures were approved by the Institutional Animal Care and Use Committee of the APQA of South Korea (approval no. 2019-462). All efforts were made to minimize animal suffering.

## 3. Results

### 3.1. Identification of Candidate Vaccine Strains and Purification of Antigens

To develop a vaccine strain capable of broad-spectrum protection, new virus strains were created using reverse genetics technology. The new virus strains JC-VP1 and PA2-VP1 were created by replacing the DNA sequence of the VP1 region of the O Manisa strain with that of O/SKR/Jincheon/2014 (SEA/Mya-98 lineage) and O/PAK/44/2008 (ME-SA/PanAsia-2 lineage), respectively (Figure 1A,B). In order to develop experimental vaccines, JC-VP1 and PA2-VP1 viruses were cultured and inactivated to purify their antigens. The 146S antigen was detected using transmission electron microscopy (Appendix A). When the inactivated and purified antigens were examined crudely using a lateral flow assay, a structural protein (SP) band was identified (Appendix A). Moreover, the genetic differences in each region for conferring neutralizing-antibody-inducing antigenicity, which is critical for FMDV [8,19], were determined. Using the O Manisa strain as a reference, 1, 2, 2, and 3 out of 29 residues were changed into O Boeun, O PA2, O TWN97, and O JC, respectively (Appendix A). Residue 191 of VP2 was different in three viruses; residue 58 of VP3 was different in two viruses; and residue 43 of VP1, residue 134 of VP3, and residue 195 of VP3 were different in one virus. The difference in the surface structure (VP1 G-H loop) between JC-VP1 and PA2-VP1 was determined through the molecular biological structural modeling of the recombinant virus capable of broad-spectrum defense (Appendix A). Mutations in the 3B region caused by replacing 3B_1_B_2_ with two iterations of 3B_3_ in the viral genome can be used as markers to distinguish vaccine viruses from wild-type strains [22,23]. 

### 3.2. Evaluation of the Protective Ability of the Experimental Vaccines in Mice

Before performing a direct experiment on pigs, a PD_50_ experiment, a protective capability test, was conducted to verify the effectiveness of the vaccines in mice, which represent an experimental animal model system. The JC-VP1 and PA2-VP1 vaccines were inoculated in 1/10, 1/40, 1/160, and 1/640 dose groups, and seven days after the vaccination, one representative virus for each of the SEA, ME-SA, and Cathay topotypes was selected and inoculated in mice for a virus challenge. The survival rate and weight change were then monitored once a week. The results of the mouse PD_50_ (mPD_50_) test revealed that the group vaccinated with JC-VP1 showed the lowest score of 18 when challenged with ME-A(O/VIT/2013), followed by 55.7 PD_50_ when challenged with SEA (O/SKR/Jincheon/2014) and 97 PD_50_ when challenged with Cathay (O/Taiwan/97). Contrary to the prediction that the score would be the highest when the mice were challenged with SEA, the highest score was observed when the mice were challenged with Cathay (Table 1 and Appendix A).

As predicted, the group vaccinated with PA2-VP1 scored the highest at > 128 mPD_50_ when challenged with ME-SA (O/VIT/2013), followed by a score of 32 when challenged with SEA (O/SKR/Jincheon/2014) and a score of 42 when challenged with Cathay (O/Taiwan/97) (Table 1 and Appendix A). In a comparison between the JC-VP1 group and the PA2-VP1 group, the mPD_50_ for SEA (O/SKR/Jincheon/2014) was higher in the JC-VP1 group than in the PA2-VP1 group, whereas the mPD_50_ for ME-SA (O/VIT/2013) was higher in the PA2-VP1 group than in the JC-VP1 group.

### 3.3. Immunogenicity Tests in Cattle and Pigs

After confirming the effectiveness of the vaccines in an experiment using mice, further experiments were conducted to evaluate immunogenicity in cattle and pigs, which were the target animals. The immunogenicity of the vaccines was examined through the virus neutralization test (VNT) and SP ELISA using blood samples drawn from cattle and pigs inoculated with the JC-VP1 and PA2-VP1 vaccines (Figure 2). In the case of the cattle, in which immune antibodies are known to form relatively well, only a single vaccination was provided, whereas the pigs were vaccinated twice (boosted) at 28 dpv. In the cattle, high levels of antibodies were formed only two weeks after inoculation of either of the vaccines, and the PA2-VP1 vaccine was particularly effective at the initial stage of inoculation (Figure 2). Conversely, after the inoculation of JC-VP1, the antibody levels continued to increase over time, and the levels of antibodies detected at 56 dpv were higher than those detected in the cattle immunized with PA2-VP1 (Figure 2A,B).

Similarly, in the cattle and pigs, antibodies rapidly formed at the early stage of inoculation (Figure 2C–E) after the inoculation of PA2-VP1. After the inoculation of JC-VP1, the antibody levels increased from 14 dpv, and after boosting, higher levels of antibodies were detected than in the pigs immunized with PA2-VP1 (Figure 2C–E).

### 3.4. Vaccine Matching Using Anti-Sera from Cattle and Pigs

Four weeks after the first vaccination with either JC-VP1 or PA2-VP1, serum samples were collected to perform a two-dimensional VNT using viruses belonging to the SEA, ME-SA, and Cathay topotypes, and the r1 values were determined. The serum samples of the animals vaccinated with JC-VP1 showed low matching scores with the ME-SA type, and the highest matching was observed in the SEA type (Appendix A).

The serum samples of the animals vaccinated with PA2-VP1 showed the lowest matching scores with the SEA type, and the highest r1 value was observed in the ME-SA type (Appendix A). These results show a pattern similar to the PD_50_ results obtained from the mouse experiment. Overall, the r1 value was higher in the pigs than in the cattle.

### 3.5. Evaluation of Neutralizing Antibodies after Vaccination in Pigs

To evaluate the post-immunization protective ability, the post-immunization and post-challenge immune antibodies were examined in the pigs (Figure 3). For the challenge inoculation, the post-immunization antibodies within four weeks of vaccination and the post-challenge antibodies were examined, and the results were found to be similar to those of the immunogenicity test (Figure 3A–C). After the challenge inoculation, the antibody levels against each of the three types of challenge viruses increased. Similar to the results of the immunogenicity test, the initial immune antibody levels in the pigs only showed a significant difference after 7 days (Figure 3A–C).

As predicted, the heterologous neutralizing antibody titer was the highest in the JC-VP1 group when challenged with SEA (O/SKR/Jincheon/2014) (Figure 3), and the highest neutralizing antibody titer was observed in the PA2-VP1 group when challenged with ME-SA (O/SKR/Boeun/2017). When Cathay (O/Taiwan/97) was inoculated, no difference was found between the JC-VP1 and PA2-VP1 groups.

In the JC-VP1 group, the homologous VN titer increased from 14 dpv, similar to the result mentioned above (Appendix A), while in the PA2-VP1 group, it increased rapidly only after 7 dpv (Appendix A). In addition, the homologous VN titers tended to be proportional according to the amount of JC-VP1 and PA2-VP1 antigens. In particular, when the JC-VP1 15 μg and 20 μg groups were compared, a significant difference was found at 2dpc (Figure 3). This result can be construed as the reason why the JC-VP1 15 μg group showed a 50% protection rate against ME-SA (O/SKR/Boeun/2017), whereas the JC-VP1 20 μg group showed a 100% protection rate.

### 3.6. Evaluation of Protectivity against Viruses of Three Different Topotypes in the Pigs

To determine the effectiveness of the vaccines for the last time, the pigs were subjected to challenge inoculation with SEA (O/SKR/Jincheon/2014), ME-SA (O/SKR/Boeun/2017), or Cathay (O/Taiwan/97) (Figure 4). In the control experiment for the challenge inoculation with the three types of viruses, sufficient virus shedding, the detection of viremia, and symptoms were observed less than two days after the challenge (Figure 4). When viremia was measured in the control group, the peak tended to appear on the second day after the pigs were challenged with the Cathay topotype virus, and the peak was detected on the fourth day after the pigs were challenged with either the SEA or ME-SA topotype. The clinical scores also showed a tendency to increase the fastest after the pigs were challenged with the Cathay type compared to the other two topotypes.

The group vaccinated with JC-VP1 showed a 100% protection rate when challenged with SEA or Cathay and a 50% protection rate when challenged with the ME-SA type (Figure 4). However, a 100% protection rate could be achieved by increasing the amount of antigen to 20 μg for the vaccination (Figure 4). The group vaccinated with PA2-VP1 showed a 100% protection rate when challenged with any of the three topotypes (Figure 4).

## 4. Discussion

Globally, about 60% of FMD outbreaks are caused by type O viruses of all FMDV serotypes [18]. Only a few vaccine strains can be widely used against type O viruses, including O1 Manisa (ME-SA/PanAsia), O 3039, O1 Campos (EURO-SA), O Tur/5/2009 (ME-SA/PanAsia2), O/IND/ R2/75, and O/Udonthani/1987 [8,24,25,26,27,28,29,30,31]. However, even these vaccines do not always match perfectly with or cannot provide complete protection against all circulating viruses [18]. Therefore, efforts should be made to obtain a vaccine strain that is a suitable match for various viruses in circulation [18,24,32].

The conventional method for preparing a vaccine strain capable of broad-spectrum protection is to select an appropriate vaccine strain from the large number of viruses available via screening in order to find a strain that has a strong enough immunogenicity to induce high antibody levels after vaccination and testing and that has an antigenicity that matches well enough with several viruses to provide a high level of protection [5,32]. However, as it is problematic to secure a large number of viruses for the selection of an appropriate vaccine strain, it is very difficult to obtain the desired vaccine strain, except for in endemic countries where FMD is persistent. We had previously attempted to develop a broad-spectrum vaccine strain that could provide protection against the SEA topotype, which belongs to the type O virus and had been circulating at the time, by replacing VP1 of the O Manisa virus (ME-SA topotype) with that of the SEA topotype [33]. This study found that antibodies capable of providing protection against the viruses of four major topotypes were induced in pigs [33]. In addition, in other studies, vaccine strains against Mya-98 and PanAsia-2 were developed, and their protective effects against viruses of the SEA topotype were examined in pigs [13].

In the present study, we further investigated how broadly the combination of the O ME-SA/PanAsia lineage and the ME-SA/PanAsia-2 lineage or the SEA/Mya-98 lineage, which are known as broad-spectrum vaccine strains, affected the vaccines’ protective abilities, and we confirmed that the new vaccines showed different levels of effectiveness. Furthermore, VP1 was found to have a significant effect on vaccine matching and protection against the wild-type virus strain.

The animal experiments using mice, pigs, and cattle revealed that ME-SA and SEA showed a tendency of vaccine matching depending on whether VP1 was composed of the same virus topotype. In the case of Cathay, when its VP1 was the SEA topotype, the protective ability in mice was relatively higher than that of ME-SA (Table 1), but antibody formation after vaccination was relatively poor despite the protective ability in pigs being acceptable when challenged with Cathay (Figure 3 and Figure 4).

It was confirmed that VP1 replacement with the SEA topotype conferred more effective protection against the challenge with viruses of the SEA topotype. The same vaccine strain showed less effective protection against the challenge with other viruses, such as ME-SA. Therefore, O Manisa, which is the standard vaccine strain for FMD, was used as a template to insert the VP1 of the Tur/05/09 (PAK44, the virus most similar to the Tur/05/09 of PanAsia2) lineage into the genome of O Manisa [24] via replacement in order to induce proper protection against all virus types, including the relatively vulnerable SEA type and other types, such as ME-SA and Cathay. The results of the challenge inoculation of the three topotypes, namely, ME-SA, SEA, and Cathay, which belong to the type O virus, showed that symptoms appeared a few days after the highest viral peak was observed in the serum, and the serum data tended to match the clinical score and the level of protection rather than the data obtained from the nasal/oral swabs (Figure 4). When challenged with ME-SA (O/Boeun/SKR/2017, O BE), FMD symptoms were observed with low VN titers, which suggests that the VN titer prevents viremia during infection, thereby acting as a major protection index.

In the case of JC-VP1, into which the VP1 of the SEA topotype was inserted, the antibody levels continuously increased after a single inoculation in the cattle. Conversely, in the case of PA2-VP1, the initial antibody induction was robust, but the ability to maintain antibody levels after 42 days was weaker than that in JC-VP1 (Figure 2).

In the PA2-VP1 vaccination groups, rapid immune formation was induced from the beginning in both the cattle and the pigs. The evaluation of protective ability conducted only four weeks after the inoculation of the pigs revealed that all immunized animals were protected, indicating that PA2-VP1 is superior to JC-VP1. Protection against Cathay was even observed at neutralizing antibody levels lower than those of the other two topotypes. This seems to be the effect of cellular immunity, but additional research is needed in this regard.

## 5. Conclusions

The PA2-VP1 vaccine strain developed in the present study was proven to provide broad-spectrum protection against all three of the tested topotypes. JC-VP1 proved to be a suitable vaccine for SEA. When applied with an increased antigen level, it could provide protection against the viruses of all three tested topotypes, making it a suitable vaccine for use in areas where SEA topotype outbreaks frequently occur. Although these vaccine strains showed little difference in their protective ability in the experimental results obtained from mice and pigs, which were used as experimental animals, they were confirmed to provide a wide range of protection. In the future, it is necessary to classify viruses with different genetic properties, even if these viruses belong to the three topotypes, and to determine precisely whether they can be protected against.

## Figures and Tables

**Figure 1 vaccines-11-00271-f001:**
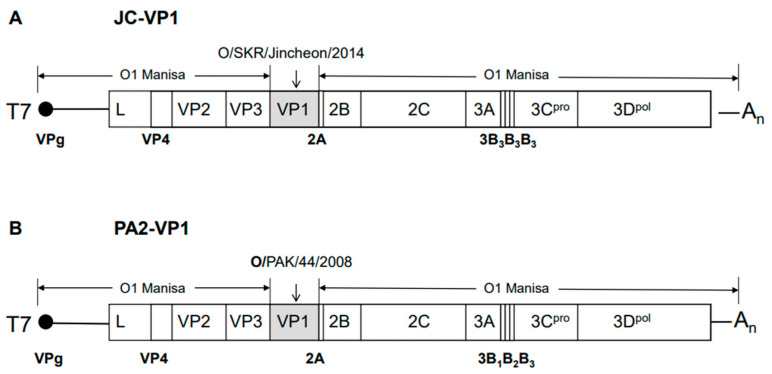
Schematic diagram of FMDV genomes with VP1 replacement for the development of a broad-spectrum vaccine strain. (**A**) JC-VP1, which has the P1 genome of O1 Manisa (VP4, 2, 3) and O/SKR/Jincheon/2014 (VP1), and (**B**) PA2-VP1, which has the P1 genome of O1 Manisa (VP4, 2, 3) and O PA2, O PAK/44/2008 (VP1). The 3B mutation of the 3B region was obtained using the same methods used in a previous report.

**Figure 2 vaccines-11-00271-f002:**
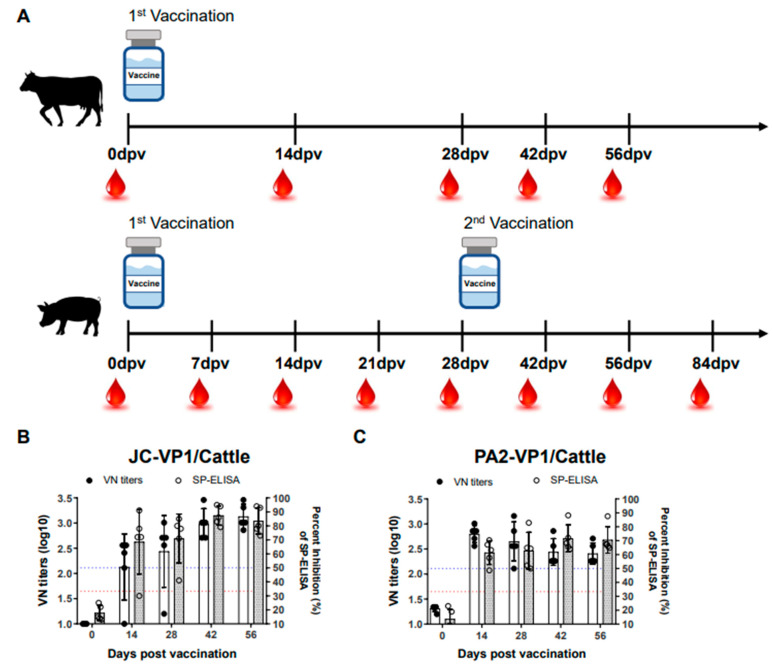
Antibody responses of cattle and pigs vaccinated with either JC-VP1 or PA2-VP1. Schematic diagram of study strategy (**A**). Neutralizing antibody titers and SP ELISA in cattle vaccinated with JC-VP1 (**B**) and PA2-VP1 (**C**). Cattle were inoculated with a single dose (15 μg/dose of inactivated antigen mixed with ISA 206, saponin, and aluminum hydroxide gel adjuvant). Neutralizing antibody titers and SP ELISA in pigs vaccinated with JC-VP1 (**D**) and PA2-VP1 (**E**). Percent inhibition (PI) > 50 was considered the cutoff of a positive reaction (blue dashed line). VN titers (log10) > 1.65 were regarded as positive (red dashed line). The pigs were inoculated with a second vaccination (15 μg/dose of inactivated antigen mixed with ISA 206, saponin, and aluminum hydroxide gel adjuvant) at 28 dpv. Comparative analysis of JC-VP1 and PA2-VP1 in vaccinated cattle and pigs (**F**). Statistical analyses were performed ANOVA and t test. ^*^
*p* < 0.05.

**Figure 3 vaccines-11-00271-f003:**
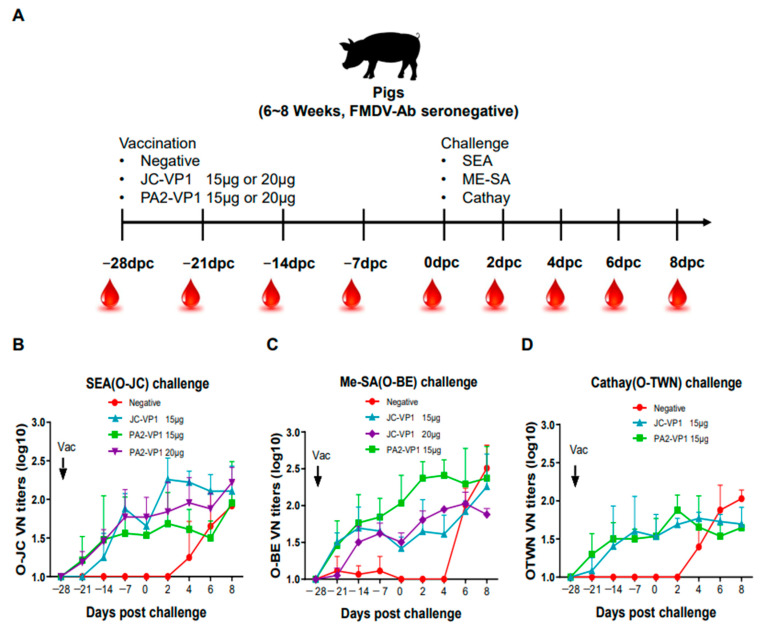
Variation in neutralizing antibody titers in pigs immunized with experimental vaccines. The pigs were inoculated with a second vaccination (15 μg/dose of inactivated antigen mixed with ISA 206, saponin, and aluminum hydroxide gel adjuvant) at 28 dpv. Schematic diagram of study strategy (**A**). The pigs were challenged with the virus of SEA (O-JC) topotype (**B**), ME-SA (O-BE) topotype (**C**), and Cathay (O-TWN) topotype (**D**) after vaccination with either JC-VP1 or PA2-VP1. In the case of PA2-VP1, different doses of 15 μg or 20 μg against the SEA topotype virus were evaluated. In the case of JC-VP1, different doses of 15 μg or 20 μg against the ME-SA topotype virus were evaluated.

**Figure 4 vaccines-11-00271-f004:**
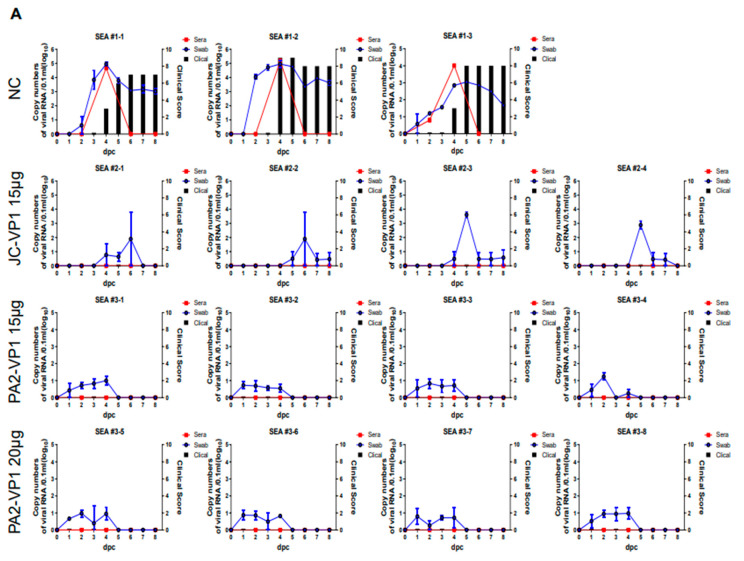
Protection in immunized pigs challenged with the viruses of the three different topotypes after vaccination with JC-VP1 or PA2-VP1. The negative control groups (#1-1, #1-2, #1-3) and vaccinated group with JC-VP1 or PA2-VP1 (#2-1, #2-2, #2-3, #2-4, #3-1, #3-2, #3-3, #3-4 were tested using a 15 μg inactivated antigen vaccine and #2-5, #2- 6, #2-7, #2-8, #3-5, #3-6, #3-7, #3-8 were tested using a 20 μg inactivated antigen vaccine) were challenged with SEA (**A**), ME-SA (**B**), and Cathay (**C**). Clinical scores (black bar), viremia (red, serum; blue, oral swab) were evaluated up to 8 days post-challenge. The JC-VP1-vaccinated groups were challenged with the viruses of the three different topotypes. Complete protection against SEA and Cathay was confirmed when the pigs were vaccinated with 15 μg of inactivated antigen vaccine of JC-VP1, but only 50% protection against ME-SA was confirmed. Complete protection against ME-SA was confirmed when tested using 20 μg of inactivated antigen vaccine of JC-VP1. The PA2-VP1-vaccinated groups were challenged with the viruses of the three different topotypes. Complete protection against the three topotype viruses was confirmed when the pigs were vaccinated with 15 μg or 20 μg of inactivated antigen vaccine of PA2-VP1.

**Table 1 vaccines-11-00271-t001:** Protection test (PD_50_) performed in vaccinated mice using 3 topotype challenge viruses.

Vaccine Strain	JC-VP1	PA2-VP1
Virus		
SEA (O/SKR/Jincheon/2014)	55.7	32.0
ME-SA (O/VIT/2013)	18.0	>128.0 ^1^
Cathay (O/Taiwan/97)	97.0	42.0

^1^ 100% survival rate was confirmed in the 1/640 dose group.

## Data Availability

All datasets generated for this study are included in the article.

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
