# Peer review of "Evaluation of Vaccine Strains Developed for Efficient, Broad-Range Protection against Foot-and-Mouth Disease Type O"

_vaccines, 2023, doi:10.3390/vaccines11020271_

Round 1
Reviewer 1 Report
Overall, this is a very interesting manuscript in which the authors present very interesting data on experimental FMDV vaccines produced using chimeric virus strains. It contributes novel data to the FMDV vaccine field through a well-designed vaccinate - challenge experiment. I find the science sound however, I think the manuscript could use significant editing to improve clarity.
Some suggestions I would offer to the authors are:
Figure #4 is very difficult to read. Some of the axis labels are too small. This figure should be edited to improve clarity. Please consider splitting this figure into multiple figures (either by challenge virus or vaccine) to make it easier to read.
Please add which animals were given a dose of 15ug of vaccine in the figure legend for Figure 4 (as you did with 20ug).
Did you perform any type of viral isolation, such as plaque assay? The included qRT-PCR data is very informative, but it would be beneficial to include data on viable virus (or lack thereof) recovered from the challenged animals.
Have you considered showing the SDG results (OD readings) in the supplemental material to show the antigen is still intact?
You mention the mutations in the 3B region in the schematic in figure 1, but I did not see any other mention of this in the manuscript. Could you provide more detail on this? Have you considered adding a marker to this vaccine that could differentiate vaccinated and infected animals by an ELISA for non-structural proteins? Please elaborate on this.
Could you please elaborate and expand on the details of the methods of the SP ELISA described in figure 2. Could you please elaborate in the figure legend of the figure 2 to include a description of dashed lines in fig 2A-D. Is this an indication of the positive/negative for the SP-ELISA? If so, you mention in lines 149-150 that: "When the samples reflected percent inhibition values of > 50 %, the animals were regarded as having demonstrated an immune response" however the dashed in the figure appears to be at 30%. Please clarify.
The right axis in figure 2 is labeled with "SP-ELISA". What the units? are you reporting percent inhibition here? Please clarify.
I would recommend adding an experiment schematic at the beginning of the paper describing when vaccination, boost, and challenge were conducted as I found the text a bit unclear at times.
An overall re-reading and editing on the manuscript would help improve clarity throughout. There are several sections in the introduction that contain run-on and unclear sentences.
Author Response
Response to Reviewer 1 Comments
We thank you for your interest in our experimental data and for your helpful comments.
Point 1: Figure #4 is very difficult to read. Some of the axis labels are too small. This figure should be edited to improve clarity. Please consider splitting this figure into multiple figures (either by challenge virus or vaccine) to make it easier to read.
Response 1: As advised, I made it look good by splitting this figure into multiple figures and increasing the figure size.
Point 2: Please add which animals were given a dose of 15ug of vaccine in the figure legend for Figure 4 (as you did with 20ug).
Response 2: I have attached information about 15ug in the figure 4 legend, please check it.
Point 3: Did you perform any type of viral isolation, such as plaque assay? The included qRT-PCR data is very informative, but it would be beneficial to include data on viable virus (or lack thereof) recovered from the challenged animals.
Response 3: It would be nice to isoloate viruses by individual, but it is difficult, and we think that it is sufficient to check viremia and protection in challenge animails using qRT-PCR.
Point 4: Have you considered showing the SDG results (OD readings) in the supplemental material to show the antigen is still intact?
Response 4: Based on your advice, we have added the OD measurement results to Supplementary Figure 1.
Point 5: You mention the mutations in the 3B region in the schematic in figure 1, but I did not see any other mention of this in the manuscript. Could you provide more detail on this? Have you considered adding a marker to this vaccine that could differentiate vaccinated and infected animals by an ELISA for non-structural proteins? Please elaborate on this.
Response 5: we thought the proportion of 3B mutations in this paper was not large, but we think it would be better to refer to the #203-205 line for readers after seeing the advice. As you said, we chose 3B mutation because we thought it would be good to use it as a marker to distinguish it from wild viruses when we started making recombinant vaccines. In a our paper published in 2020, there is a result that the wild-type virus and the 3B mutation virus are easily distinguished with lateral flow assay (PBM co. ltd) kit.
Point 6: Could you please elaborate and expand on the details of the methods of the SP ELISA described in figure 2. Could you please elaborate in the figure legend of the figure 2 to include a description of dashed lines in fig 2A-D. Is this an indication of the positive/negative for the SP-ELISA? If so, you mention in lines 149-150 that: "When the samples reflected percent inhibition values of > 50 %, the animals were regarded as having demonstrated an immune response" however the dashed in the figure appears to be at 30%. Please clarify.
Response 6: Thank you for pointing out the confusion due to the lack of explanation for the dashed line. A schematic diagram was added for the figure 2 experimental method. we explained the dashed line in the figure. Blue is the positive cutoff for ELISA and red is the positive cutoff for VNT.
Point 7: The right axis in figure 2 is labeled with "SP-ELISA". What the units? are you reporting percent inhibition here? Please clarify.
Response 7: we modified The right axis to percent inhibition to increase the clarity.
Point 8: I would recommend adding an experiment schematic at the beginning of the paper describing when vaccination, boost, and challenge were conducted as I found the text a bit unclear at times.
Response 8: Based on what you mentioned, we added a schematic diagram of the experiment in Figures 2a and 3a.
Point 9: An overall re-reading and editing on the manuscript would help improve clarity throughout. There are several sections in the introduction that contain run-on and unclear sentences.
Response 9: we tried to improve the clarity by modifying the introduction part.
Reviewer 2 Report
Dear Authors
The sentence structures of the paper are difficult in making any sense. Therefore, you are requested to change them. e.g., line #57-62. Line # 70 states "To this end, vaccine strains with broad antigenic" makes no sense. kindly rewrite this line also. Methods section will be re written once again and altogether because it is completely plagiarized. Figures are fine and are easy to understand. Introduction is too long and needs size reduction. Overall, the paper has unrationalized self-citation that must be avoided. Figure legends in supplementary materials appear to be copied as such from somewhere. The authors are requested to critically reanalyze the complete contents of this research paper before proceeding further.
Author Response
Response to Reviewer 2 Comments
We thank you for your interest in our experimental data and for your helpful comments.
Point 1: The sentence structures of the paper are difficult in making any sense. Therefore, you are requested to change them. e.g., line #57-62. Line # 70 states "To this end, vaccine strains with broad antigenic" makes no sense. kindly rewrite this line also.
Response 1: we tried to increase the clarity of the introduction part, including line #57-62. Line #70.
Point 2: Methods section will be re written once again and altogether because it is completely plagiarized.
Response 2: As you advised, we tried to rewrite the method part as a whole.
Point 3: Introduction is too long and needs size reduction.
Response 3: we made a slight reduction by revising the introduction.